# Novel Therapeutic Target Critical for SARS-CoV-2 Infectivity and Induction of the Cytokine Release Syndrome

**DOI:** 10.3390/cells12091332

**Published:** 2023-05-07

**Authors:** William W. Harless, Beth Lewis, Bessi Qorri, Samar Abdulkhalek, Myron R. Szewczuk

**Affiliations:** 1ENCYT Technologies Inc., Membertou, NS B1S 0H1, Canada; 2Department of Biomedical and Molecular Sciences, Queen’s University, Kingston, ON K7L 3N6, Canada; 3Faculty of Health Sciences, Higher Colleges of Technology, Abu Dhabi P.O. Box 25026, United Arab Emirates

**Keywords:** SARS-CoV-2, angiotensin-converting enzyme 2, cytokine release syndrome, neuraminidase-1, Toll-like receptors

## Abstract

We discovered a novel therapeutic target critical for SARS-CoV-2, cellular infectivity and the induction of the cytokine release syndrome. Here, we show that the mammalian enzyme neuraminidase-1 (Neu-1) is part of a highly conserved signaling platform that regulates the dimerization and activation of the ACE2 receptors and the Toll-like receptors (TLRs) implicated in the cytokine release syndrome (CRS). Activated Neu-1 cleaves glycosylated residues that provide a steric hindrance to both ACE2 and TLR dimerization, a process critical to both viral attachment to the receptor and entry into the cell and TLR activation. Blocking Neu-1 inhibited ACE2 receptor dimerization and internalization, TLR dimerization and activation, and the expression of several key inflammatory molecules implicated in the CRS and death from ARDS. Treatments that target Neu-1 are predicted to be highly effective against infection with SARS-CoV-2, given the central role played by this enzyme in viral cellular entry and the induction of the CRS.

## 1. Introduction

Severe acute respiratory syndrome coronavirus 2 (SARS-CoV-2) spike S protein binds to the host cell membrane receptor protein angiotensin-converting enzyme 2 (ACE2) [1]. The biochemical mechanism triggering ACE-2 receptor dimerization and subsequent permissive state allowing for viral binding to the ACE2 homodimer and cellular internalization is unknown. Steric hindrance from peptidase domain (PD) glycosylated residues attached to the surface membrane receptor can interfere with the dimerization of the ACE2 receptor in preventing receptor activation and cellular entry [2]. These surface receptors are abundant on both pulmonary epithelial cells and macrophages [3]. Infection and activation of pulmonary macrophages are hypothesized to be the critical driver of a host hyper-immune response described as a “cytokine storm” and defined as the cytokine release syndrome (CRS) [4]. Toll-like receptors (TLR) play a central role in this viral-induced uncontrolled immune response that can result in death from acute respiratory distress syndrome (ARDS) [5].

TLRs play critical roles in a host’s response to various infectious organisms [6]. They are activated by bacteria and viruses’ pathogen-associated molecular patterns (PAMPs) [7]. This response triggers the innate immune response, with the production of inflammatory cytokines and the indirect activation of the adaptive immune response. This response activation is usually adaptive; however, when dysregulated, it can cause the host’s death from a dysregulated immune response that damages tissue and vital organ function [8].

The betacoronavirus SARS-CoV-2 has been found to activate many TLRs implicated in the cytokine storm and ARDS [9]. However, the precise mechanism(s) for this activation remains unclear. The SARS-CoV-2 spike protein S1 subunit can trigger an inflammatory response via TLR-4 signaling in murine and human macrophage cell lines [10]. TLR-2 can also sense the SARS-CoV-2 envelope protein to produce inflammatory cytokines implicated in the cytokine storm [11]. TLRs have been linked to SARS-CoV-2-induced CRS [12].

Neuraminidases are responsible for the cleavage of sialic acid groups from carbohydrates [13]. These sialic acid groups are often closely associated with various cellular membrane receptors and play critical roles in receptor dimerization, activation, and signaling [14]. We have previously reported an unprecedented mechanism for pathogen molecule-induced TLR activation dependent on neuraminidase-1 (Neu-1) sialidase activity [15]. We showed a membrane-controlling mechanism initiated by ligand binding to cell surface TLR-4, -7 and -9 [15,16,17,18] to induce Neu-1 sialidase activity in live primary bone marrow (BM) macrophage cells, and macrophage and dendritic cell lines. Central to this process is that Neu-1 and not Neu-2, -3, or -4 forms a complex with TLR-2, -3, and -4 in combination with the neuromedin B G-protein coupled receptor (GPCR) and matrix metalloproteinase-9 (MMP-9) on the cell surface of naïve macrophages. Infectious signals such as lipopolysaccharide (LPS) binding to the TLR will facilitate GPCR conformational change and activation of MMP-9 with subsequent activation of Neu-1. Neu-1 cleaves α-2,3 sialic acid residues, removing steric hindrance for receptor dimerization, activation and subsequent downstream signaling with the activation of NFκB and nitric oxide and pro-inflammatory IL-6 and TNFα cytokines production in macrophage cell lines [19]. This tripartite neuromedin B GPCR-MMP-9-Neu-1 signaling paradigm appears to be highly conserved when we discovered the same signaling mechanism regulating tyrosine kinase receptors (RTKs) such as epidermal growth factor (EGF) [20], insulin receptor (IRβ) [21,22,23,24], and neurotrophin Trk [25,26,27,28].

The interesting question is whether Neu1 neuraminidase plays a role in regulating cell surface receptors critical for SARS-CoV-2 viral cellular entry and the induction of the CRS. It is reported that the betacoronavirus SARS-CoV-2 infects human cells using the spike (S) glycoprotein binding to the angiotensin-converting enzyme receptor-2 (ACE2) [29]. During the infection, the S glycoprotein is cleaved into subunits, S1 and S2. The subunit S1 contains the receptor-binding domain (RBD), which allows the virus to bind directly to the peptidase domain (PD) of ACE2. S2 plays a role in facilitating membrane fusion and subsequent translocation of the virus and receptor into the cell. The ligand(s) triggering the ACE2 receptor and subsequent permissive state allowing for viral binding to the ACE2 homodimer are unknown. Steric hindrance from PD glycosylated residues attached to the surface membrane receptor can interfere with the dimerization of the ACE2 receptor in preventing receptor activation and cellular entry [2]. Cryo-electron microscopy structures of full-length human ACE2 reveal the presence of the neutral amino acid transporter B0AT1 (B0AT1) with or without the receptor binding domain (RBD) of the surface S protein of SARS-CoV-2 at the ACE2-RBD interface. The ACE2-B0AT1 complex is assembled as heterodimers, with the collectrin-like domain of ACE2 mediating homodimerization. The resultant homodimer has two PDs, which can bind two SARS-CoV-2 S protein trimers simultaneously. The glycosylation moieties are located within the PD structures [30]. The glycosylation moieties are proposed to prevent ACE2 dimerization. Reports from protein crystallography studies, sequence analyses of ligand binding to extracellular domains of ACE2, and receptor dimerization analyses strongly conclude that ACE2 dimerization is an essential step in the receptor activation mechanism [30]. Although the ligand-binding and the receptor dimerization steps are critical for ACE2 activation, the molecular mechanism(s) behind these two events remains unknown.

Given our previous work showing the critical role of Neu-1 in the activation of specific TLRs and the hypothesized role of glycosylated residues inhibiting ACE2 receptor dimerization and a permissive state for viral entry, we further explore in this study whether Neu-1 may play a role in SARS-CoV-2 pathogenesis and its potential as a therapeutic target in the treatment of SARS-CoV-2 infections.

## 2. Material and Methods

### 2.1. Cell Lines

RAW-Blue™ cells (Mouse Macrophage Reporter Cell Line, InvivoGen, San Diego, CA, USA) derived from RAW 264.7 macrophages are grown in a culture medium containing Zeocin as the selectable marker. They stably express a secreted embryonic alkaline phosphatase (SEAP) gene inducible by NF-κB and AP-1 transcription factors. Upon stimulation, RAW-Blue™ cells activate NF-κB and AP-1, leading to the secretion of SEAP, which is detectable and measurable using QUANTI-Blue™, a SEAP detection medium (InvivoGen). RAW-Blue™ cells are resistant to Zeocin™ and G418 in the conditioned medium. NCI-H1299 (ATCC, CRL-5803™, carcinoma; non-small cell lung cancer) is an epithelial-like cell isolated from a White, 43-year-old male patient with carcinoma. A549 (ATCC, CCL-185™, lung carcinoma) is an epithelial-like lung cell isolated from the lung tissue of a White, 58-year-old male with lung cancer. Cells were maintained in DMEM supplemented with 10% fetal bovine serum (FBS) and 0.1% plasmocin and grown in T-75 tissue culture flasks up to 70–80% confluence prior to experimental use.

### 2.2. Ligands and Reagents

TLR4 ligand, lipopolysaccharide (LPS) from BioVision was used at 50 μg/mL. Neuropeptide FF, bradykinin acetate salt, and angiotensin I-7 human acetate salt (all purchased from Sigma-Aldrich) diluted in Tris-buffered saline (TBS, pH 7.4) were used at indicated concentrations. In the live cell sialidase assay experiments, 2-(4-methylumbelliferyl)-α-D-N-acetylneuraminic acid (98% pure 4-MUNANA; Biosynth International Inc., Itasca, IL, USA) is a sialidase substrate used at a pre-determined concentration of 0.318 mM diluted in TBS. Recombinant spike His-tag proteins were obtained from R&D Systems a Biotechne brand with the following samples: Alpha variant (UK, SARS-CoV-2 B.1.1.7 N501Y Spike His-tag Protein, CF, Catalog # 10748-CV-100), Beta variant (S. Africa, SARS-CoV-2 B.1.351 Spike RBD His Protein, CF, Catalog # 10735-CV-100), Delta variant (India, SARS-CoV-2 B.1.617.2 Spike GCN4-IZ Protein, CF, Catalog # 10878-CV-100), Gamma variant (Brazil, SARS-CoV-2 P.1 Spike GCN4-IZ His-tag Protein, CF, Catalog # 10795-CV-100), Kappa variant (India, SARS-CoV-2 B.1.617.1 Spike GCN4-IZ Protein, CF, Catalog # 10861-CV-100), BA.2 variant (omicron, SARS-CoV-2 BA.2 Spike (GCN4-IZ) His Protein, CF, Catalog # 11109-CV-100), BA.4 and BA.5 variants (omicron, SARS-CoV-2 BA.4/BA.5 Spike His-tag Protein, CF, Catalog # 11232-CV-100). Omicron variant (recombinant SARS-CoV-2 B.1.1.529 spike S1 protein, His-tag, Catalog # 40589-V49H6-B) obtained from Sino Biological US Inc., 10101 Southwest Freeway, Suite 100. Houston, TX 77074, USA).

### 2.3. Inhibitors

Acetylsalicylic acid (ASA, >99% pure, Sigma–Aldrich, Steinheim, Germany) was dissolved in dimethyl sulfoxide (DMSO) to prepare a 550 mM stock solution, which was stored in aliquots at −20 °C. The highest-used aspirin concentration contains less than 0.5% *v*/*v* of DMSO in 1× PBS at a pH of 7. Oseltamivir phosphate-USP (OP, batch No. MBAS20014A, >99% pure powder, Solara Active Pharma Sciences Ltd., New Mangalore-575011, Karnataka, India) was freshly dissolved in sterile normal saline before use. Peramivir (Pera, lot # 23765-1(CA), Cedarlane Labs 4410 Paletta Court, Burlington, ON L7L 5R2) was freshly dissolved in sterile normal saline before use. MMP-9 inhibitor (MMP9i, Calbiochem-EMD Chemicals Inc., Darmstadt, Germany) is a cell-permeable, potent, selective, and reversible MMP-9 inhibitor (IC50 = 5 nM). It only inhibits MMP-1 (IC50 = 1.05 μM) and MMP-13 (IC50 = 113 nM) at much higher concentrations. BIM-46174 is a selective inhibitor of the heterotrimeric G-protein complex. BIM-46174 prevents the heterotrimeric G-protein signaling linked to several GPCRs mediating (a) cyclic AMP generation (Gαs), (b) calcium release (Gαq), and (c) cancer cell invasion by Wnt-2 frizzled receptors and high-affinity neurotensin receptors (Gαo/i and Gαq). BIM-46174 was kindly provided by IPSEN Innovation (91940 Les Ulis, France). BIM-23127 is a specific neuromedin B receptor inhibitor from Tocris Bioscience (Tocris House, IO Centre Moorend Farm Avenue, Bristol, BS11 0QL, United Kingdom).

### 2.4. NF-κB-Dependent Secretory Alkaline Phosphatase (SEAP) Assay

Briefly, a cell suspension of 1 × 10^6^ cells/mL in the fresh growth medium was prepared, and 100 μL of RawBlue cell suspension (∼100,000 cells) was added to each well of a Falcon flat-bottom 96-well plate (Becton Dickinson). Different concentrations of either specific MMP9 inhibitor (MMP9i), oseltamivir phosphate (OP), BIM-23127 (BIM23), BIM-46174 (BIM46) ot peramivir (Pera) were added to each well 1 h. before ligand stimulation. The plates were incubated at 37 °C in a 5% CO_2_ incubator for 18–24 h. A QUANTI-Blue™ (InvivoGen) solution, a detection medium developed to determine the activity of any alkaline phosphatase present in a biological sample, was prepared following the manufacturer’s instructions. Briefly, 160 μL of resuspended QUANTI-Blue solution was added to each well of a flat-bottom 96-well plate, followed by 40 μL of supernatant from stimulated RAW-blue cells. The plate was incubated for 60 min at 37 °C, and the SEAP levels were determined using a spectrophotometer at 620–655 nm. Each experiment was performed in triplicates.

### 2.5. Neu-1 Colocalization with ACE2

RawBlue macrophage cells, H1299 epithelial-like human lung cell line and the A549 human lung adenocarcinoma cell line were cultured in a DMEM medium with 10% FCS and 0.1% plasmocin. Cells were treated with 10 μg/mL of SARS-CoV-2 variant spike S proteins and left untreated as controls. Cells were fixed, permeabilized and immunostained with mouse anti-ACE2 conjugated with Alexa Fluor594 (ACE2 E-11, sc-390851 AF594, Santa Cruz Biotechnology, Inc., Dallas, TX, USA) and mouse anti-Neu-1 conjugated with Alexa Fluor488 (Neu-1 F-8 sc-166824 AF488, Santa Cruz Biotechnology, Inc.). Stained cells were visualized using a Zeiss M2 fluorescent microscope imager with a 20× or 40× objective. To calculate the percentage of colocalization in the selected images, the Pearson correlation coefficient was measured on a total of 15–20 cells per image and expressed as a percentage using Image J 1.38x software (NIH, USA).

### 2.6. Live Cell Microscopy

Live cell microscopy imaging was performed using an inverted microscope (Leica DMi8) equipped with a Photron Fastcam SA-Z high-speed camera. RAW-blue cells and A549 human lung cells (50,000 cells) were plated on a 35 mm MatTek dish (MatTek Headquarters, 200 Homer Ave Ashland, MA 01721, USA), No. 1.5 gridded coverslip at 14 mm glass diameter in culture media containing 10% fetal calf sera for 24 h, and left untreated. Live cells were washed with 1× PBS and immunostained with mouse anti-ACE2 conjugated with Alexa Fluor594 and mouse anti-Neu-1 conjugated with Alexa Fluor488. Stained live cells were visualized using an inverted Leica DMi8 microscope equipped with a Photron Fastcam SA-Z high-speed camera with 100× oil objectives. The kinetic expression of ACE2 (red) and Neu-1 (green) colocalization were taken every second and recorded as a video.

### 2.7. Mouse Cytokine ELISA Array III (Colorimetric) Sandwich Assay

RawBlue macrophage cells were maintained in DMEM supplemented with 10% fetal bovine serum (FBS) and 0.1% plasmocin and grown in T-75 tissue culture flasks up to 70–80% confluence prior to experimental use. Cells at 50,000 were plated in 24 well tissue culture for 24 h. After the media was removed, the cells were pretreated with 1× DMEM (without supplements) containing LPS (50 μg/mL), omicron (6.25 μg/mL), or delta (6.25 μg/mL) with indicated 800 μg/mL OP plus 10 mM ASA or left untreated as control (bkg). The supernatants were centrifuged to clear any debris and 100 μL were added to each well of the cytokine ELISA array plate coated with 32 different antibodies against indicated mouse cytokines for 2 h. at room temperature and mixed on a rocker plate. The test sample is allowed to react with a pair of antibodies, resulting in the cytokines being sandwiched between the solid phase and enzyme-linked antibodies. After incubation, the wells are washed to remove unbound-labeled antibodies. To the plate was added streptavidin conjugated horse radish peroxidase (HRP) and substrate TMB (3,3′,5,5′-tetramethylbenzidine). The level of expression for each specific cytokine is directly proportional to the luminescent or color intensity. The results are expressed as fold change of the treated cells to untreated controls.

### 2.8. MILLIPLEX^®^ Luminex^®^ 200™ xMAP^®^ Flow Cytometry-Based Instrumentation

MILLIPLEX^®^ products are based on the Luminex^®^ xMAP^®^ technology. Luminex^®^ products use proprietary techniques where microspheres are internally color-coded with multiple fluorescent dyes and prepared using precise concentrations into magnetic polystyrene microspheres, each of which is coated with a specific capture antibody. Since multiple conjugated beads may be added to one sample, multiple results may be obtained from each sample. During testing, when an analyte attaches to the bead and a biotinylated detection antibody is added, followed by Streptavidin-Phycoerythrin conjugate, the reaction is completed and the plate is run on the analyzer. The Luminex^®^ 200™ analyzer is a flow cytometry-based instrument that quantifies the fluorescent reporter signals for each individual microsphere. The INTELLIFLEX software generates data that may be exported in xPONENT^®^ style CSV files. Prior to testing, the analyzer was calibrated and performance verification procedure were performed successfully.

RawBlue macrophage cells were maintained in DMEM supplemented with 10% fetal bovine serum (FBS) and 0.1% plasmocin and grown in T-75 tissue culture flasks up to 70–80% confluence prior to experimental use. Cells at 50,000 were plated in 24 well tissue culture for 24 h. After the media was removed, the cells were pretreated with 1× DMEM (without supplements) containing LPS (50 μg/mL), beta, gamma, kappa, alpha, UK, BA.2 or omicron (6.25 μg/mL) with indicated 800 μg/mL OP plus 10 mM ASA or left untreated as control (bkg). Seventeen tissue culture supernatants were prepared by Dr MR Szewczuk in the Laboratory at Queen’s University and shipped on ice to Encyt Technologies where they were frozen at −80 °C on receipt. On the day of testing, the samples were thawed, vortexed, and then centrifuged for 1 min at 14,000 rpms to ensure samples were free of particles. Samples were prepared for testing on the Luminex^®^ 200™ following the procedure in the MILLIPLEX^®^ Mouse Cytokine/Chemokine Magnetic Panel User Guide for the selected cytokines (Kit Catalog# Hcytomag-70K, Lot#3893941, MilliporeSigma, St. Louis, MO, USA.). The kit and components were stored at 2–8 °C until the day of testing. The cytokines tested in this kit are G-CSF, MIP-1A, and MCP-1.

The matrix solution, 1× DMEM medium without serum, that was added to the background, standards, and controls during set up, which was the media used in the preparation of the samples. It had been shipped with the samples and was stored at 4 °C until the day of testing. After overnight incubation at 2–8 °C on a plate shaker set at 775 rpms, the well contents were gently removed and washed ×2 with wash buffer using the handheld magnet. Detection antibodies were added to the wells and incubated at room temperature on a plate shaker for one hour. Streptavidin-phycoerythrin was then added to each well and incubated on a plate shaker at room temperature for 30 min. Well contents were then gently removed and the wells washed ×2 with wash buffer using the handheld magnet. xMAP Sheath Fluid Plus which is the delivery medium that carries the sample to the optics component of Luminex xMAP technology-based instruments. was added to each well, mixed on the plate shaker for 5 min, and immediately run on the Luminex^®^ 200™.

### 2.9. Supplementary Material for Live Cell Video

Live cell microscopy using an inverted microscope (Leica DMi8) equipped with a Photron Fastcam SA-Z high-speed camera for RawBlue macrophage cells immunostained with mouse anti-ACE2 conjugated with Alexa Fluor594 and mouse anti-Neu-1 conjugated with Alexa Fluor488.

### 2.10. Statistical Analysis

Data are presented as the mean ± the standard error of the mean (SEM) from at least three repeats for each experiment performed in triplicate. Comparisons between two groups from three independent experiments were made by one-way analysis of variance (ANOVA) at 95% confidence using the uncorrected Fisher’s LSD multiple comparisons test with 95% confidence with asterisks denoting statistical significance.

## 3. Results

### 3.1. Neu-1 Sialidase Activity Is Activated by SARS-CoV-2 Spike Protein S1 Stimulation in Murine Macrophages and Human Epithelial Cells

Similar to the role of Neu-1 in activating TLRs by cleaving alpha 2,3 sialic acid residues and removing steric hindrance to receptor dimerization and activation, we propose a similar molecular link regulating the interaction and signaling mechanism(s) between the SARS-CoV-2 S protein and the TLRs implicated in the cytokine storm and the ACE-2 receptor. Central to this process is that NMBR-MMP-9-Neu-1crosstalk is tethered at the ectodomain of ACE2 receptors on the cell surface (Figure 1A).

We first investigated if the SARS-CoV-2 virus S protein and its variants trigger Neu-1 activity. As shown in Figure 1B–H, all variants (alpha, beta, delta, gamma, omicron, BA.2, BA.4 and BA.5) of SARS-CoV-2 spike S proteins and endotoxin lipopolysaccharide (LPS) were able to induce sialidase activity. Due to the role of NMBR in inducing MMP9 elastase activity and subsequent Neu-1 enzymatic activity, we hypothesized that NMBR inhibition using BIM-46174 and BIM-23127 would suppress Neu-1 sialidase activity in GPCR agonist-stimulated and S protein-stimulated RawBlue cells. BIM-23127 is a specific inhibitor of NMBR. BIM-46174 is a selective inhibitor of heterotrimeric G-protein complex, preventing the heterotrimeric G-protein signaling linked to several GPCRs mediating (a) cyclic AMP generation (Gαs), (b) calcium release (Gαq), and (c) cancer cell invasion by Wnt-2 frizzled receptors and high-affinity neurotensin receptors (Gαoi and Gαq). BIM-46174 (BIM46), as shown in Figure 1B,C, significantly inhibited Neu-1 sialidase activity induced by recombinant S1 spike protein alpha (UK) in RawBue cells. BIM-23127 (BIM23) also significantly inhibited the S protein activation of Neu-1 sialidase (Figure 1F,G). The S1 spike protein was used at 20 μg/mL, and BIM46174 and BIM-23127 at 200 μg/mL. We also used an MMP-9 inhibitor (MMP-9i), a cell-permeable, potent, selective, and reversible inhibitor (IC_50_ = 5 nM). It can also inhibit MMP-1 (IC_50_ = 1.05 μM) and MMP-13 (IC_50_ = 113 nM) only at much higher concentrations. MMP9i, as shown in Figure 1B,F,G, significantly inhibited Neu-1 sialidase activity induced by S proteins from SARS-CoV-2 alpha (UK), BA.4 and BA.5 in RawBue cells.

Notably, the sialidase activity associated with S protein stimulation of live RawBlue cells is attributed to Neu-1 and not the other three mammalian neuraminidases [21]. Those studies found that only anti-Neu-1 antibodies blocked the sialidase activity associated with insulin-treated cells comparable to the untreated controls [31]. In contrast, antibodies against the other three enzymes, Neu-2, -3 or -4, had no inhibitory effect on sialidase activity associated with insulin stimulation [21]. The same study also used human WG544 or 1140F01 type 1 sialidosis fibroblast cell lines, genetically deficient in Neu-1. It showed no significant induction of Neu-1 sialidase activity in insulin-stimulated Neu-1-deficient cells compared to the WT human fibroblast cells [21]. Another study used similar GPCR agonists to induce Neu-1 sialidase activity in primary bone marrow macrophages from the hypomorphic cathepsin A mice with a 90% reduction of the Neu-1 activity (Neu-1–CathA KD) [17].

Similarly, GPCR agonist stimulation of Neu-1-deficient macrophages did not significantly induce Neu-1 sialidase activity in vitro. These data suggest that Neu-1 sialidase is primarily activated in the presence of stimulatory RTK ligands such as insulin and GPCR agonists, and S1 spike proteins are consistent with these previous reports [15,16,17,18,21]. Here, oseltamivir phosphate (OP), acetylsalicylic acid (ASA), and peramivir (pera) significantly inhibited the sialidase activity induced by any of the variant S proteins (Figure 1C–H).

### 3.2. Mammalian Neu-1 Is Tethered to Naive ACE2 Receptors

According to the cryo-electron microscopy structures of full-length human ACE2 in the presence of the neutral amino acid transporter B^0^AT1 with the RBD of the surface S protein of SARS-CoV-2 (Figure 2A), homodimerization is entirely mediated by the ACE2, which is sandwiched by B^0^AT1 [29]. Both the PD and neck domains contribute to ACE2 dimerization, and each B^0^AT1 interacts with the neck and transmembrane (TM) helix in the adjacent ACE2. The extracellular region is highly glycosylated, with seven and five glycosylation sites on each ACE2 and B^0^AT1 monomer, respectively [29]. We speculated that the ACE2 glycosylation sites might play a crucial role in the homodimerization following SARS-CoV-2 S glycoprotein binding. Other reports have suggested that receptor glycosylation modification may be the connecting link between ligand binding, receptor dimerization and activation for several other receptors [20,32,33,34].

For example, growth factors binding to their RTKs utilize GPCR-signaling molecules to initiate the molecular organizational-signaling platform of a novel Neu-1 and MMP-9 crosstalk in alliance with RTK on the cell surface. We have identified this novel GPCR-signaling platform essential for neurotrophin factor-induced Trk [25], insulin [21,22,23,24], epidermal growth factor (EGF) [20], and TLR [15,16,17] receptors activation processes and downstream cellular signaling. These findings have revealed a novel concept in which GPCR activation is essential for the growth factor and pathogen activation of TLR. The mechanism(s) here involves the formation of a functional signaling complex between the GPCR-RTK and GPCR-TLR partners. To test whether this same organizational signaling platform is involved with ACE2 activation, we initially asked if Neu-1 forms a complex with ACE2 receptors.

As depicted in Figure 2A, the glycosylation moieties of the ACE2 receptor are shown as sticks [29]. The complex is colored by subunits, with the PD and C-terminal collectrin-like domain (CLD) in one ACE2 protomer colored cyan and blue, respectively. Here, we hypothesize that the GPCR-MMP-9-Neu-1 signaling platform is tethered to the ACE2 at the ectodomain near the CLD, as illustrated in Figure 2B. The premise is that the glycosylation modification of the side chains of the PD and CLD subunits is critical for the ACE2 activation associated with SARS-CoV-2 S protein binding to the RBD. Because of the glycosylation patterns of ACE2 receptors, it is proposed that Neu-1 and MMP-9 crosstalk on the cell surface is an essential requirement for regulating the activation of ACE2 receptors.

This crosstalk discloses a novel receptor-signaling paradigm involving an agonist-induced GPCR-signaling process via Gα proteins and MMP-9 activation in inducing Neu-1 sialidase, which forms a tripartite complex with ACE2 at the ectodomain on the cell surface (Figure 2B). It is proposed that S-protein binding ACE2 causes a conformation change of ACE2 to induce GPCR signaling to potentiate MMP-9 activation. MMP-9, or gelatinase B, is one of the largest and most complex members of the MMP family of at least 28 members. MMP-9 can bind to gelatin, collagens type I, IV, V, and other substrates. The elastin-degrading activity of MMP-9 fits well within this novel signaling paradigm of ACE2 activation. This process involves the elastin-binding protein (EBP), which is part of the molecular multi-enzymatic complex that contains β-galactosidase/Neu-1 and protective protein cathepsin A. Direct removal of EBP from the complex by activated MMP-9 is proposed to activate Neu-1.

In support of the signaling model of ACE2, the data shown in Figure 2C,D validated the predicted association of Neu-1 and ACE2. Zeiss M2 Imager fluorescence microscopy revealed the membrane colocalization of Neu-1 (anti-Neu-1 conjugated with Alexa Fluor 488) with ACE2 (anti-ACE2 conjugated with Alexa Fluor 594) in naive RAW-blue cells. There was no reduction of Neu-1 colocalization with ACE2 in these naïve cells. It is noteworthy that after adding the recombinant S proteins from alpha (Figure 2E), beta (Figure 2F), delta (Figure 2G) and kappa (Figure 2H) SAR-CoV-2 variants S proteins for 5 min, the percentage of ACE2-Neu-1 colocalization was reduced to zero. These data indicate that the ACE2 receptor binding to the S proteins may rapidly internalize into the cell.

To confirm these observations, live cell microscopy and kinetic video expression of ACE2 and Neu-1 colocalization were assessed using an inverted Leica DMi8 fluorescent microscope equipped with a photron fastcam SA-Z high-speed camera. Live RawBlue and A549 human lung cells were cultured on a 35 mm Mattek dish, No. 1.5 gridded coverslip at 14 mm glass diameter in culture media containing 10% fetal calf sera for 24 h and left untreated. Live cells were washed with 1× PBS and immunostained with mouse anti-ACE2 conjugated with Alexa Fluor 594 and mouse anti-Neu-1 conjugated with Alexa Fluor 488. The data depicted in Figure 3A,B reveal live kinetic expression per second of ACE2-Neu-1 colocalization on one cell. Surprisingly, there is an increased expression of ACE2-Neu-1 expression of three receptors at 1 s, eleven at 4 s, and eighteen at 6 s for RawBlue macrophages. Similar results were found with A549 human lung cells (Figure 3C). Live cell videos of RawBlue and A549 are in Appendix A for Figure 3B and Appendix A for Figure 3D, respectively.

It is noteworthy that H1299 epithelial-like human lung cell and the A549 human lung adenocarcinoma cell line express ACE2-Neu-1 colocalization as well (Figure 4A,B). We used the A549 cell line to ask whether SAR-CoV-2 variant S proteins would impact the ACE2-Neu-1 colocalization. Here, we measured the Pearson correlation coefficient, which measures the strength of a linear association between two variables. A Pearson product-moment correlation attempts to draw a line of best fit through the data of two variables. A high Pearson value represents the best-fit correlation between the two variables. The data depicted in Figure 4C reveal a variation in the ACE2-Neu-1 Pearson coefficient in response to the delta, kappa, gamma and beta SARS-CoV-2 S proteins compared to the untreated A549 cells.

The data reveal a differential response of ACE2-Neu-1 colocalization to the different SARS-CoV-2 S proteins. For example, following OP treatment in Figure 4C, the kinetics of colocalization with the S protein together with OP was 10 min. To further examine this issue, we did a kinetic study up to 30 min of treatment (Figure 4D); here, the data reveal that delta S protein in combination with OP after 20 min and up to 30 min showed an increase in the Pearson colocalization coefficient, indicating higher expression of ACE2-Neu-1 expression on the cell surface. These data confirm that OP treatment blocks ACE2 internalization into the cell.

### 3.3. Recombinant SARS-CoV-2 S Protein Induces Upregulation of NF-kB That Is Downregulated by Selective Inhibitors of Neu-1 Oseltamivir Phosphate (OP) and Aspirin (ASA)

To better study the mechanism of the induction of the cytokine storm triggered by SARS-CoV-2 and the possible role of Neu-1 in this process, we investigated the effect of S proteins, LPS, bradykinin, Ang1-7, and neuropeptide FF for their ability to upregulate nuclear factor-κB (NF-κB). NF-kB represents a family of transcription response profiles. Here, we used RAW-Blue™ cells (mouse macrophage reporter cell line). These cells stably express a secreted embryonic alkaline phosphatase (SEAP) gene inducible by NF-kB and activator protein-1 (AP-1) transcription factors. AP-1, a transcription factor, regulates gene expression in response to various stimuli, including cytokines, growth factors, stress, and bacterial and viral infections. Upon stimulation, RAW-Blue™ cells activate NF-kB and AP-1, leading to the secretion of SEAP, which is detectable and measurable when using QUANTI-Blue™, a SEAP detection medium.

If Neu-1 activity is associated with GPCR-signaling and MMP-9 activation in live TLR-expressing macrophage cells, we asked if GPCR agonists binding to their respective GPCR receptors would directly induce sialidase-activity in the absence of TLR specific ligand. The data in Figure 5 are consistent with this hypothesis. Here, GPCR agonists Ang1-7 (Mas GPCR), neuropeptide FF (NPFFR1 GPCR), and bradykinin (BR1 GPCR) heterodimerize with bombesin neuromedin B receptor (NMBR) to induce Neu-1 sialidase activity in live RawBlue macrophage cells compared to LPS in a dose-dependent manner. Interestingly, the different SARS-CoV-2 variants S proteins induced marked sialidase activity compared to LPS (Figure 5H,I). These responses were inhibited by OP, aspirin (ASA) and peramivir (pera), dose-dependently.

### 3.4. Recombinant SARS-CoV-2 S Protein Stimulation of RawBlue Macrophages Triggers the Upregulation of Distinct Cytokines Implicated in SARS-CoV-2 Pathogenesis

Cytokine storm syndrome can lead to ARDS, a primary cause of mortality in COVID-19 disease. Therefore, therapies that suppress the cytokine storm are essential for preventing disease deterioration in COVID-19 patients and saving patients’ lives, which is significant for treating critically ill patients and reducing the mortality rate. Here, we investigated the cytokine and chemokine responses to recombinant SARS-CoV-2 S protein (omicron or delta variants) stimulation of the mouse RawBlue macrophage cell line and the modulatory effect of OP and ASA on distinct cytokines and chemokines production. Figure 6 shows an increasing trend of Tumor Necrosis Factor (TNF)-α, insulin-like growth factor (IGF), Vascular Endothelial Growth Factor (VEGF), IL6, basic Fibroblast Growth Factor (FGF-b), Interferon (IFN)-γ, Epidermal Growth Factor (EGF), Leptin, IL1α, IL1β, Granulocyte-Colony Stimulating Factor (G-CSF), Granulocyte–Macrophage Colony Stimulating Factor (GM-CSF), Monocyte Chemoattractant Protein (MCP)-1, Macrophage Inflammatory Protein 1-alpha (MIP-1α), Stem cell factor (SCF), Regulated on Activation Normal T-cell Expressed and Secreted (RANTES/CCL5), Platelet-derived growth factor (PDGF)-BB, basic nerve growth factor (b-NGF), IL-17a, IL-2, IL-4, IL-10, Resisten, IL-12, CCL11, CCL21, IL-3, IL-13, IL-121, IL-22, CXCL10, CXCL1, and upon RawBlue cells stimulation with omicron or delta variants S proteins. Pre-treatment with OP and ASA was able to reduce the production of particular cytokines and chemokines in response to omicron S protein stimulation that includes IGF-1, IL-6, FGF-b, INF- γ, EGF, Leptin, G-CSF, GM-CSF, MCP-1, MIP-1α, SCF, Rantes, PDGF-bb, b-NGF, IL-17α, IL-2, IL-4, IL-12, IL-13, IL-21 and IL-22 (Figure 6B). The same pre-treatment with OP and ASA had less effect on cytokine production upon the delta S protein stimulation, and the reduction in cytokines was seen in TNF-α, IGF-1, IL-6, INF- γ, EGF, MIP-1 α, SCF, Rantes, b-NGF, and IL-10 but not the others (Figure 6C).

G-CSF, MCP-1 and MIP1-α are chemokines that are involved in the rapid recruitment of immune cells and the activation of macrophages, which were shown to be increased in patients with ARDS [35]; therefore, we tested the effect of OP plus ASA treatment on G-CSF, MCP-1 and MIP1-α production in response to different SARS-CoV-2 variants stimulation (Figure 7). OP plus ASA pre-treatment significantly reduced all three chemokines secretion in response to alpha (UK) and BA.2 (omicron subvariant), similar to the omicron. Notably, these inhibitory effects were significant, with a reduction in G-CSF and MIP-1A of approximately 99% (Figure 7).

## 4. Discussion

Worldwide deaths from SARS-CoV-2 infection are estimated at approximately 5.5 million [36]. Death is typically caused by the pathophysiologic process known as ARDS triggered by a hyper-immune response to the virus [37]. This hyper-immune response, or cytokine storm, is induced by macrophage activation and the release of distinct cytokines predictably upregulated in patients suffering from ARDS [38]. Therapies developed to reduce this loss of life include vaccination to prevent infection and ameliorate its effects with medications to reduce the damaging effects of active viral infection. Most vaccines and treatments approved to date are designed to target the S protein. While these therapies have proven somewhat effective in reducing the rate of symptomatic SARS-CoV-2 infection and death [39,40], variants resistant to current vaccinations and treatments continue to develop because of SARS-CoV-2 high infectivity and replication rates [41]. Mutations in the coding region for the S protein are the primary determinant of resistance to most current treatments [42,43]. While more virulent SARS-CoV-2 variants resistant to current treatments are an ongoing concern, perhaps an even graver concern is the possible re-emergence of variant coronaviruses with much higher case fatality rates. For example, SARS-CoV-1 and Middle East respiratory syndrome (MERS) had case fatality rates of 10% [44] and 35% [45], respectively. Like SARS-CoV-2, SARS-CoV-1 infects human cells through the interaction of the S protein with the ACE2 receptor [46].

We provide evidence that both the ACE2 receptor and specific TLRs implicated in SARS-CoV-2 pathophysiology are regulated by mammalian Neu-1. Neu-1 and MMP-9 form a crosstalk signaling platform with the NMBR GPCR for both ACE2 and TLR-2, -3, -4 [15,19], and TLR-7 and -9 [18]. S-protein binding to ACE2 and the TLRs will cause a conformational change in these receptors, triggering GPCR signaling to activate MMP-9 and subsequent activation of Neu-1. Notably, Neu-1 can also be activated by this same tripartite signaling complex with other GPCR agonists such as Ang1-7 (Mas GPCR), neuropeptide FF (NPFFR1 GPCR), and bradykinin (BR1 GPCR) by heterodimerizing with the bombesin neuromedin B receptor (NMBR) tethered to the ACE2 and TLR receptors (Figure 5) to induce Neu-1 sialidase activity. Activated Neu-1 desialylates the α-2,3 sialyl residues of ACE2 and the TLR leading to receptor dimerization/activation. ACE2 receptor dimerization will allow the virus to bind tightly to the receptor and become internalized. In contrast, TLR dimerization will trigger the activation of the inflammatory cascade that can lead to CRS, clinical deterioration, and patient death from ARDS.

Blocking Neu-1 inhibited S protein-induced ACE2 receptor dimerization/internalization and the expression of critical cytokines implicated in the CRS and the pathophysiology associated with ARDS. We also observed that by blocking Neu-1, the ACE2 receptor repopulation induced by receptor dimerization/internalization could be reduced significantly (Figure 4). These results are significant critical determinants of ACE2–SARS-CoV-2 RBD interactions [47]. However, accelerated ACE2 receptor repopulation following internalization promoted by spike protein binding to the ACE2 receptor has not been recognized until now to be a possible critical determinant of SARS-CoV-2 pathogenicity (Figure 3).

Intriguingly, some recent clinical evidence in humans supports the hypothesis that targeting Neu-1 may prove effective in treating SARS-CoV-2 infections. In a retrospective study, critically ill hospitalized patients with COVID-19 pneumonia treated with neuraminidase inhibitors oseltamivir or peramivir had a significantly reduced fatality rate compared to the control group [48]. After propensity score-matched analysis, the mortality rate in the neuraminidase inhibitor-treated group (28 out of 486 subjects, 5.8%) was significantly lower than that in the control group (61 out of 486 subjects, 12.6%, *p* < 0.001) (5.8% vs. 12.6%). This is equivalent to a hazard ratio of 0.46, which compares favorably with the currently approved standard of care treatment in patients hospitalized with COVID-19 pneumonia [49,50]. The authors acknowledged that they did not understand why these treatments proved effective, given that the SARS-CoV-2 virus does not contain neuraminidase. However, we think that the results from this study can be explained at the cellular level by the ability of these medications to inhibit mammalian Neu-1, disrupting viral entry into the cell and the release of cytokines triggering the CRS and ARDS, as we have demonstrated in our experiments (Figure 1C–H) and a potential therapeutic target not currently recognized.

Treatment with aspirin has also been explored as a possible treatment for COVID-19 infections. There is evidence that aspirin use can reduce the risk of infection from SARS-CoV-2, reduce hospital mortality, and prevent clinical deterioration in those most at risk for dying [51,52]. A plausible mechanism for aspirin’s potential effectiveness in treating SARS-CoV-2 infections is the discovery that aspirin has been reported to inhibit Neu-1 sialidase as well [53]. In support of this research, Yang et al. [54] show that host Neu1 regulates coronavirus replication by controlling sialylation on coronavirus nucleocapsid protein.

In our experiments, it is noteworthy with caution that treatment with Neu-1 inhibitors reduced the expression of specific cytokines implicated in the CRS and ARDS by three orders of magnitude (Figure 7). While this effect is predicted to be highly beneficial in patients suffering from CRS and ARDS, it could be potentially harmful as it could also disrupt an adaptive immune response of the host to the virus. The significant benefit Neu-1 inhibitors demonstrated in the study referenced above should provide some reassurance to their further exploration in clinical trials, as well as the realization that people dying from SARS-CoV-2 infection are dying from an unregulated immune response leading to the CRS and ARDS.

Since Neu-1 is involved in several cellular activities, the possible adverse effects of treatments that target the Neu-1 enzyme are abdominal or stomach cramps or tenderness; arm, back, or jaw pain; bloating; chest pain or discomfort; diarrhea, watery; drooling; facial swelling; and fast or irregular heartbeat. The OP drug that was used in our study is a widely utilized medication with a known side effect profile and has been shown to be relatively safe. The drug has also been shown to be relatively safe with few side effects when given intravenously to healthy volunteers.

## 5. Conclusions

We provide evidence in this paper that a highly conserved molecular signaling platform central to SARS-CoV-2 pathophysiology can be disrupted by targeting Neu-1. Treatments that target the Neu-1 enzyme have the potential to be highly effective against this virus by limiting not only the cytokine storm but also preventing viral entry into the cell. It is unlikely that such a highly conserved signaling platform can be easily overcome by ongoing viral selection and mutation. Targeting Neu-1 provides an opportunity for effective treatment at the current time and in the future, with the predicted emergence of novel SARS-type viruses resistant to current vaccination and treatments that target the S protein.

## Figures and Tables

**Figure 1 cells-12-01332-f001:**
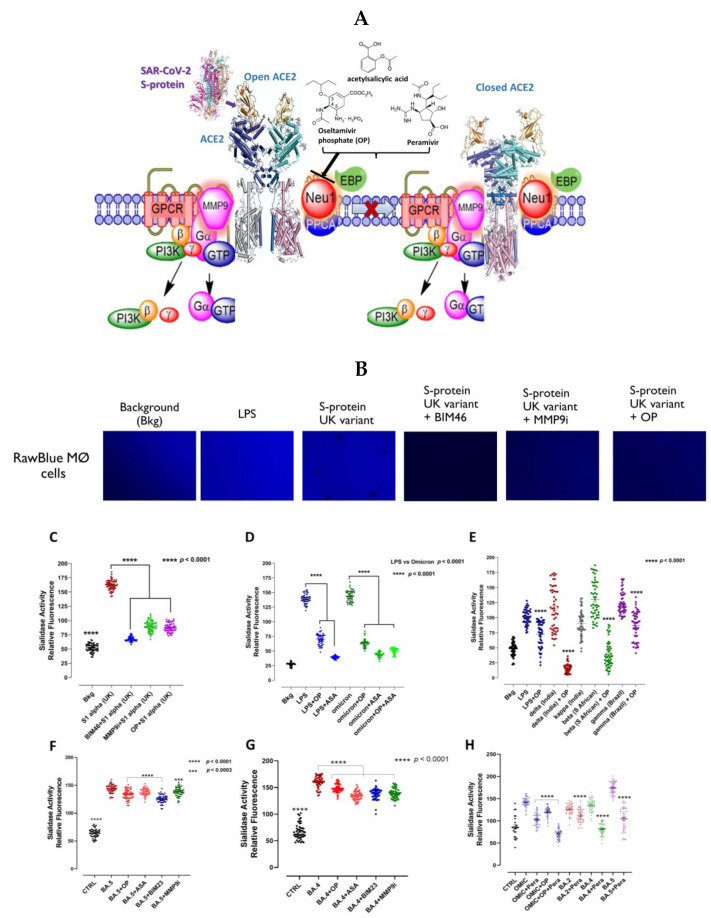
(**A**) We propose that the ACE2 receptor exists in a multimeric receptor complex with neuromedin-B GPCR receptor, ACE2, and Neu1 in naïve (unstimulated) and stimulated ACE2-expressing cells. Here, a novel molecular signaling platform regulating the interaction and signaling mechanism(s) between these molecules on the cell surface uncovers an S protein induced ACE2 activation signaling axis mediated by Neu1 sialidase activation and the modification of ACE2 PD glycosylation. This signaling platform potentiates neuraminidase-1 (Neu1) and matrix metalloproteinase-9 (MMP-9) crosstalk on the cell surface, which is essential for activating ACE2. Notes: SAR-CoV-2 S protein triggers ACE2 conformational change to potentiate NMBR-ACE2 signaling and MMP-9 activation to induce Neu1 sialidase. Activated MMP-9 is proposed to remove the elastin-binding protein (EBP) as part of the molecular multi-enzymatic complex that contains β-galactosidase/Neu1 and protective protein cathepsin A (PPCA). Activated Neu1 hydrolyzes α-2,3 sialyl residues at the ectodomain of ACE2 to remove steric hindrance to facilitate ACE2 subunits association, activation, and subsequent cellular entry. Oseltamivir phosphate (OP), acetylsalicylic acid and peramivir inhibit Neu1 sialidase activity. Abbreviations: PI3K: phosphatidylinositol 3-kinase; GTP: guanine triphosphate; IRS1: insulin receptor substrate-1; p: phosphorylation. Citation: Taken partly from Haxho et al. Cellular Signalling Volume 43, March 2018, Pages 71–84. © 2023 Alghamdi et al., Published by Elsevier Inc., Open access under CC BY-NC-ND license. This Open Access article permits unrestricted non-commercial use, provided the original work is properly cited. (**B**–**H**) RawBlue macrophage cells were allowed to adhere to 12 mm circular glass slides for 24 h at 37 °C in a humidified incubator. After removing the media, 0.318 mM 4-MUNANA substrate in Tris-buffered saline pH 7.4 was added to cells alone (background, Bkg) or with indicated S-proteins 20 µg/mL or endotoxin LPS. In combination with the indicated S-proteins and 200 μg/mL oseltamivir phosphate (OP), 200 μg/mL peramivir (Pera), 200 μg/mL BIM-46174 (BIM46, inhibitor of the heterotrimeric Gα/Gβ/Gy protein complex), 200 μg/mL BIM-23127 (BIM23, a specific neuromedin B receptor inhibitor), 200 µg/mL MMP9 specific inhibitor (MMP9i) or 12 mM aspirin (ASA), the sialidase expression was markedly reduced. Fluorescent images were taken at 2 min after adding substrate using epi-fluorescent microscopy (40× objective). The mean fluorescence surrounding the cells for each image was measured using Image J Software. The results are depicted as a scatter plot of data point visualization using dots to represent the fluorescence values (*n* = 50) obtained from one representative experiment of 3 separate experiments with similar results. The relative fluorescence values of each group were compared to the indicated group by ANOVA using the uncorrected Fisher’s LSD multiple comparisons test with 95% confidence with indicated asterisks for statistical significance.

**Figure 2 cells-12-01332-f002:**
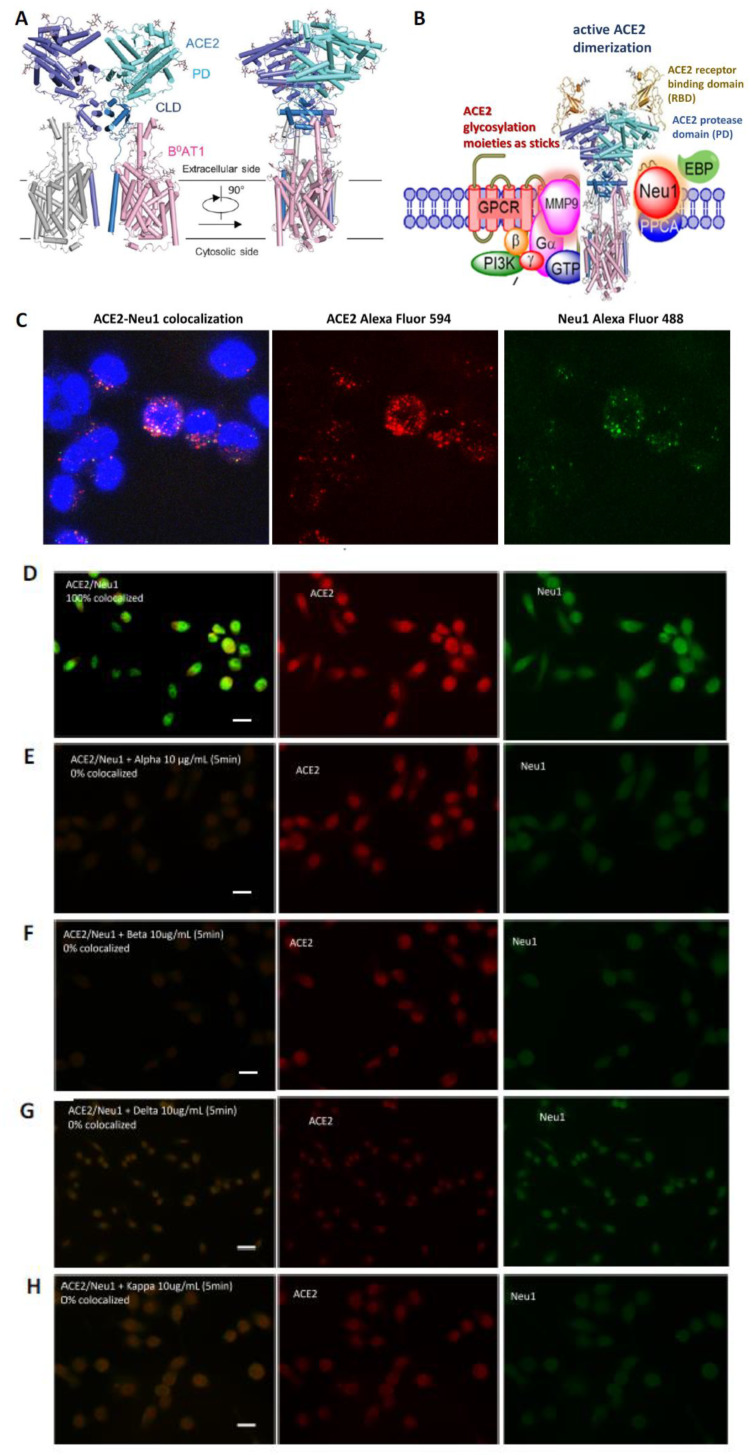
(**A**) Adapted cartoon representation of the atomic model of the ACE2-B0AT1 complex [29]. The glycosylation moieties are shown as sticks. The complex subunits are colored, with the PD and CLD in one ACE2 protomer colored cyan and blue, respectively (**B**,**C**). Neu-1 (anti-Neu-1 conjugated with Alexa Fluor 488) and ACE2 (anti-ACE2 conjugated with Alexa Fluor 594) colocalize with naïve RAW-blue cells. Cells (50,000 cells) were plated on 12 mm circular glass slides in culture media containing 10% fetal calf sera for 24 h and left untreated as controls. The data represent one of three independent experiments showing similar results. (**D**–**H**) RAW-blue cells (50,000 cells) were plated on 12 mm circular glass slides in culture media containing 10% fetal calf sera for 24 h and treated with 10 μg/mL of recombinant S1 spike protein from alpha (**E**), beta (**F**), delta (**G**) and kappa (**H**) for 5 min. Cells were fixed, permeabilized and immunostained with primary antibodies conjugated with Alex Fluor as in (**C**). Stained cells were visualized using a Zeiss M2 imager with a 20× objective. The scale bar represents 100 μm.

**Figure 3 cells-12-01332-f003:**
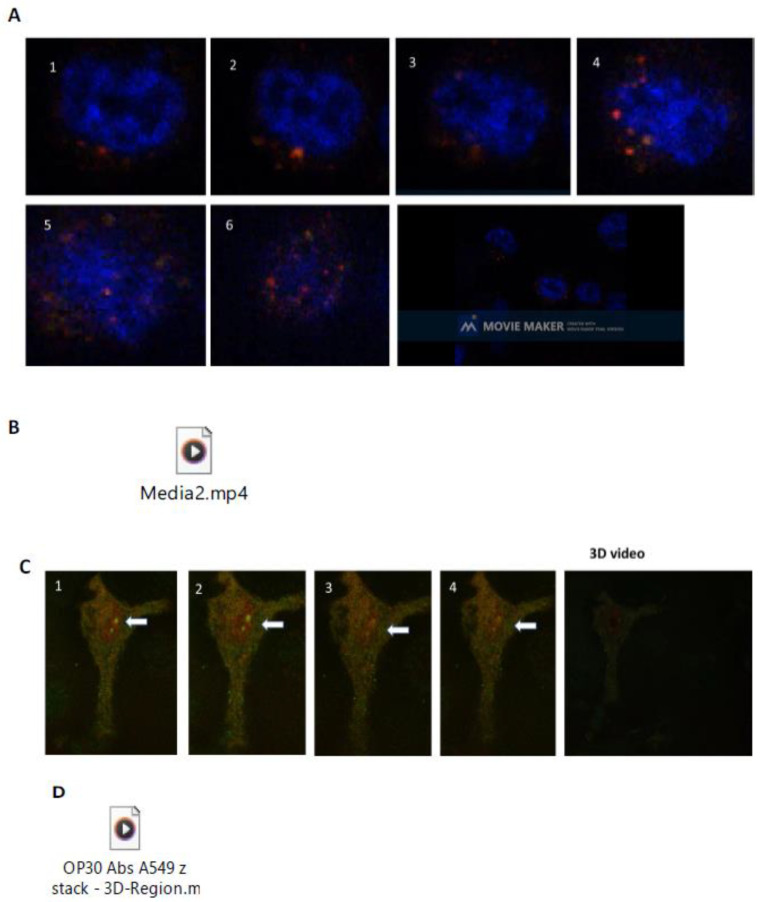
Live cell microscopy using an inverted microscope (Leica DMi8) equipped with a Photron Fastcam SA-Z high-speed camera. (**A**,**B**) RAW-blue cells and (**C**,**D**) A549 human lung cells (50,000 cells) were plated on a 35 mm MatTek dish, No. 1.5 gridded coverslip at 14 mm glass diameter in culture media containing 10% fetal calf sera for 24 h and left untreated. Live cells were washed with 1× PBS and immunostained with mouse anti-ACE2 conjugated with Alexa Fluor 594 and mouse anti-Neu-1 conjugated with Alexa Fluor 488. Stained live cells were visualized using an inverted Leica DMi8 microscope equipped with a Photron Fastcam SA-Z high-speed camera with 100x oil objectives. The kinetic expression of ACE2 (red) and Neu-1 (green) colocalization were taken every second and recorded as a Appendix A (**B**). (**C**) A549 human lung cells (50,000 cells) were plated on a 35 mm Mattek dish, No. 1.5 gridded coverslip at 14 mm glass diameter in culture media containing 10% fetal calf sera for 24 h and left untreated. Cells were stained as described in (**B**). The kinetic expression of ACE2 (red) and Neu-1 (green) colocalization for A549 cells was taken every second and recorded as a Appendix A (**D**).

**Figure 4 cells-12-01332-f004:**
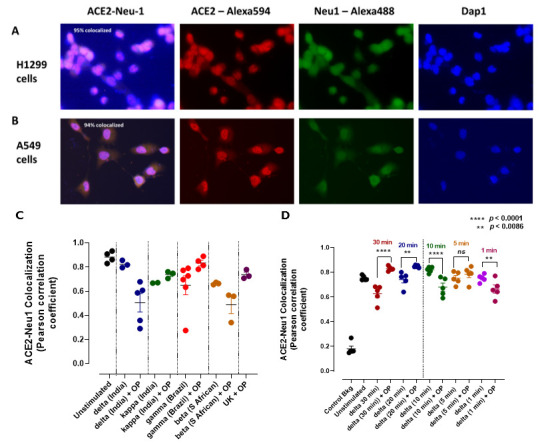
Neu-1 colocalizes with ACE2 (**A**) H1299 epithelial-like human lung cells and (**B**) human A549 lung adenocarcinoma cells. Cells (50,000 cells) were plated on 12 mm circular glass slides in culture media containing 10% fetal calf sera for 24 h and treated with 10 μg/mL delta, kappa, gamma, and beta variant S-proteins (**C**) for indicated times (**D**) with or without OP (40 mg/mL) or left untreated as controls. Cells were fixed, permeabilized and immunostained with mouse monoclonal anti-Neu-1 conjugated with Alexa488 and mouse monoclonal anti-ACE2 conjugated with Alexa594 antibodies. Stained cells were visualized using a Zeiss M2 imager fluorescent microscope with a 40× objective. The Pearson correlation coefficient was measured on a total of 5 different images per cell and expressed as a percentage using Zeiss M2 imager software to calculate the percentage of colocalization in the selected images. The data represent one of three independent experiments showing similar results.

**Figure 5 cells-12-01332-f005:**
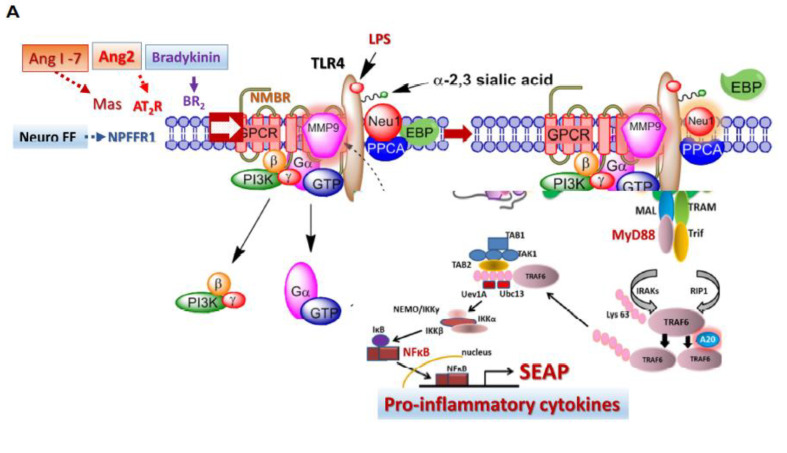
(**A**) Infographics of the RAW-Blue™ cells (mouse macrophage reporter cell line) were grown in a culture medium under zeocin selection. The cells stably express a secreted embryonic alkaline phosphatase (SEAP) gene inducible by NF-kB and AP-1 transcription factors. Upon stimulation, RAW-Blue™ cells activate NF-kB and AP-1, leading to the secretion of SEAP, which is detectable and measurable when using QUANTI-Blue™, a SEAP detection medium (Invivogen). (**B**) Quantitative spectrophotometry analysis of the effect of Ang1-7 pre-treatment on sialidase activity in live RawBlue cells. Significance is a nonparametric trend test in dose responses to test whether the response values increase or decrease with increasing dosage with indicated *p* values (*n* = 4). (**C**) Ang1-7-induced SEAP activity in the culture medium was assessed using QUANTI-Blue substrate. Relative SEAP activity was calculated as fold change of each compound (SEAP activity in medium from treated cells minus no cell background over SEAP activity in medium from untreated cells minus background). (**D**,**E**) Similar analyses were done with neuropeptide FF as with Ang1-7. (**F**,**G**) Relative SEAP activity for bradykinin and LPS. (**H**,**I**) Relative SEAP activity induced by various indicated spike proteins from SARS-CoV-2 variants compared to LPS. (**J**) Relative SEAP activity induced by alpha (UK) spike protein inhibited by OP dose-dependently. (**K**) Inhibition of omicron-induced SEAP activity with indicated doses of OP plus ASA. (**L**) Inhibition of omicron-induced SEAP activity with indicated individual doses of OP and ASA. (**M**,**N**) Inhibition of BA.4 and BA.5-induced SEAP activity with indicated individual doses (µg/mL) of peramivir (pera).

**Figure 6 cells-12-01332-f006:**
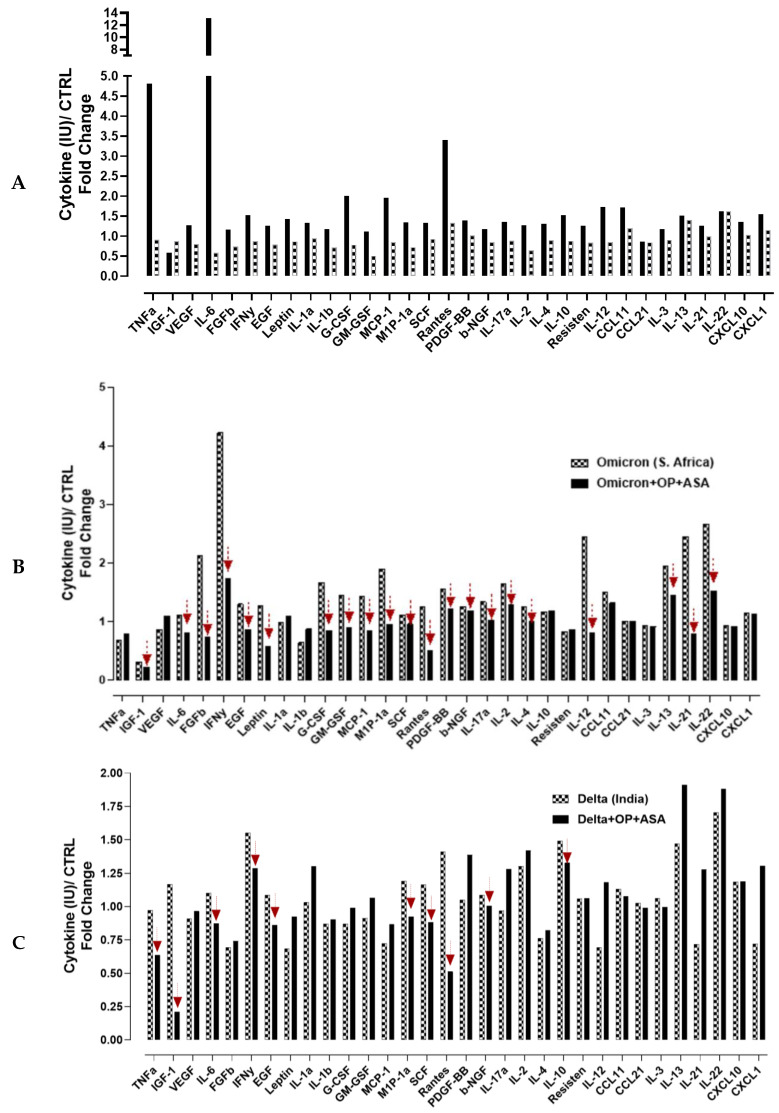
Mouse cytokine ELISA array III (colorimetric) sandwich assay to measure 32 mouse cytokines simultaneously in the supernatants of RawBlue mouse macrophages treated for three days with 50 µg/mL LPS and 6.25 µg/mL omicron S protein (**A**), omicron plus 800 µg/mL OP and 10 mM ASA treatment (**B**), and 6.25 µg/mL delta plus 800 μg/mL OP and 10 mM ASA (**C**) or left untreated as a control. The supernatants were centrifuged to clear any debris, and 100 μL were added to each well of the cytokine ELISA array coated with 32 different antibodies against indicated mouse cytokines for 2 h. at room temperature and mixed on a rocker plate. Cells at 50,000 were plated in 24 well tissue culture plates for 24 h. After the media was removed, the cells were pretreated with 1× DMEM (without supplements) containing LPS (50 µg/mL), omicron (6.25 µg/mL), or delta (6.25 µg/mL) with indicated 800 µg/mL OP plus 10 mM ASA or left untreated as control (bkg). The cell culture supernatants are incubated in the cytokine ELISA microarray plate, and the captured cytokine proteins are subsequently detected with a cocktail of biotinylated detection antibodies. The test sample is allowed to react with a pair of antibodies, resulting in the cytokines between the solid phase and enzyme-linked antibodies. After incubation, the wells are washed to remove unbound-labelled antibodies. To the plate was added streptavidin conjugated horse radish peroxidase (HRP) and substrate TMB (3,3′,5,5′-tetramethylbenzidine). The expression level for each specific cytokine is directly proportional to the luminescent or color intensity. The results are expressed as a fold change of the treated cells to untreated controls.

**Figure 7 cells-12-01332-f007:**
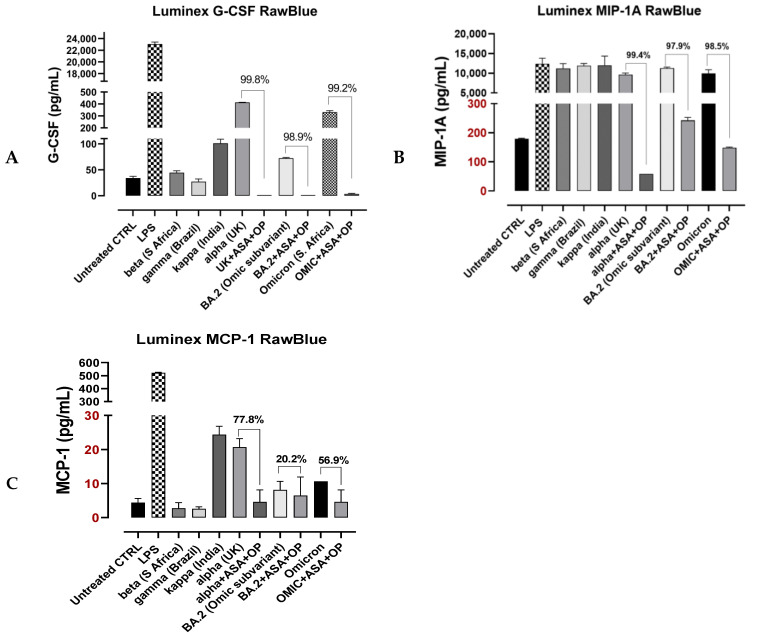
MILLIPLEX^®^ products are based on the Luminex^®^ xMAP^®^ technology. Luminex^®^ products use proprietary techniques where microspheres are internally color-coded with multiple fluorescent dyes and prepared using precise concentrations into magnetic polystyrene microspheres, each coated with a specific capture antibody. RawBlue macrophage cells were maintained in DMEM supplemented with 10% fetal bovine serum (FBS) and 0.1% plasmocin and grown in T-75 tissue culture flasks up to 70–80% confluence prior to experimental use. Cells at 50,000 were plated in 24 well tissue cultures for 24 h. After the media was removed, the cells were pretreated with 1× DMEM (without supplements) containing LPS (50 μg/mL), beta, gamma, kappa, alpha, UK, BA.2 or omicron (6.25 μg/mL) with indicated 800 μg/mL OP plus 10 mM ASA or left untreated as control (CTRL). Luminex cytokine assay to measure G-CSF (**A**), MIP-1A (**B**) and MCP-1 (**C**). The percentages indicate the inhibition of cytokine following OP plus ASA treatment.

## Data Availability

All data needed to evaluate the paper’s conclusions are present. The preclinical data sets generated and analyzed during the current study are not publicly available but from the corresponding author upon reasonable request. The data will be provided following the review and approval of a research proposal Statistical Analysis Plan and execution of a Data Sharing Agreement. The data will be accessible for twelve months for approved requests, with possible extensions considered. For more information on the process or to submit a request, contact szewczuk@queensu.ca.

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
