# Peer review of "Novel Therapeutic Target Critical for SARS-CoV-2 Infectivity and Induction of the Cytokine Release Syndrome"

_cells, 2023, doi:10.3390/cells12091332_

Round 1
Reviewer 1 Report
In the present study, the authors explored the impact of targeting intracellular Neu1 on severe acute respiratory syndrome coronavirus 2 (SARS-CoV-2) infection. This study deals one of the main research topic of this period and it responds to the need to comprehend the SARS-CoV-2 pathogenicity mechanisms and the host response to develop COVID-19 treatments and SARS-CoV-2 antivirals.
The authors, on RAW-blue cells, A549 human lung cells and H1299 epithelial-like human lung cells,
provided evidences about a negative regulatory role or the mammalian Neu-1 in S protein-induced ACE2 receptor dimerization and internalization and also in the expression of pivotal cytokines involved in CRS. Moreover, they showed that the ACE2 receptor repopulation could be reduced by blocking Neu-1.
The central role of Neu-1 in viral cellular entry and in the induction of CRS is well proved in this study and supported by well-organized and described methods. Moreover, cartoons are really useful to better visualize and comprehend the mechanism of interaction of viral S-protein and the host cell membrane protein complexes.
Live cell microscopy allows to better visualize the ACE2 and Neu-1 colocalization, but, on the contrary, images in fig 1B were not so clear. Indeed, I was not able to detect any recognizable fluorescence in any of the images, neither in the online version, nor in the printed one. I suggest the authors to adjust these images to be more visible.
Moreover, authors could discuss how their results fit with some new data recently published, as for example “Targeting intracellular Neu1 for Coronavirus Infection Treatment Darong Yang, Yin Wu, Isaac Turan, Joseph Keil, Kui Li, Michael H Chen, Runhua Liu, Lizhong Wang, Xue-Long Sun, Guo-Yun Chen bioRxiv 2022.09.09.507342; doi: https://doi.org/10.1101/2022.09.09.507342”.
Author Response
Reviewer # 1:
Comments and Suggestions for Authors
1. In the present study, the authors explored the impact of targeting intracellular Neu1 on severe acute respiratory syndrome coronavirus 2 (SARS-CoV-2) infection. This study deals one of the main research topic of this period and it responds to the need to comprehend the SARS-CoV-2 pathogenicity mechanisms and the host response to develop COVID-19 treatments and SARS-CoV-2 antivirals. The authors, on RAW-blue cells, A549 human lung cells and H1299 epithelial-like human lung cells, provided evidences about a negative regulatory role or the mammalian Neu-1 in S protein-induced ACE2 receptor dimerization and internalization and also in the expression of pivotal cytokines involved in CRS. Moreover, they showed that the ACE2 receptor repopulation could be reduced by blocking Neu-1. The central role of Neu-1 in viral cellular entry and in the induction of CRS is well proved in this study and supported by well-organized and described methods. Moreover, cartoons are really useful to better visualize and comprehend the mechanism of interaction of viral S-protein and the host cell membrane protein complexes. Author response: Thank you for the kind comments.
2. Live cell microscopy allows to better visualize the ACE2 and Neu-1 colocalization, but, on the contrary, images in fig 1B were not so clear. Indeed, I was not able to detect any recognizable fluorescence in any of the images, neither in the online version, nor in the printed one. I suggest the authors to adjust these images to be more visible. Author response: Wrong figure 1B, it is a sialidase assay. The colocalization data are in Figure 2.
3. Moreover, authors could discuss how their results fit with some new data recently published, as for example “Targeting intracellular Neu1 for Coronavirus Infection Treatment Darong Yang, Yin Wu, Isaac Turan, Joseph Keil, Kui Li, Michael H Chen, Runhua Liu, Lizhong Wang, Xue-Long Sun, Guo-Yun Chen bioRxiv 2022.09.09.507342; doi: https://doi.org/10.1101/2022.09.09.507342”.
Author response: Thank you for the comment. We have included the following to cite Yang et al. "In support of this research, Yang et al. [54] show that host Neu1 regulates corona-virus replication by controlling sialylation on coronavirus nucleocapsid protein. They found that the coronavirus nucleocapsid proteins in COVID-19 patients and in corona-virus HCoV-OC43-infected cells were heavily sialylated. Notably, this sialylation of the nucleocapsid proteins controlled the RNA-binding activity and replication of coronavirus. Neu1 overexpression increased HCoV-OC43 replication, while Neu1 knockdown reduced HCoV-OC43 replication [54]. Moreover, a newly developed Neu1 inhibitor, Neu5Ac2en-OAcOMe, selectively targeted intracellular sialidase, which dramatically re-duced HCoV-OC43 and SARS-CoV-2 replication in vitro and rescued mice from HCoV-OC43 infection-induced death [54]. These findings and our results in this report suggest that specific targeted Neu1 inhibitors could be used to limit SARS-CoV-2 replica-tion in patients with COVID-19, making Neu1 a potential therapeutic target for COVID-19 and future coronavirus pandemics."
Reviewer 2 Report
I have read with great interest the article "Novel Therapeutic Target Critical for SARS-CoV-2 Infectivity and Induction of the Cytokine Release Syndrome". I have found a well thought out idea, excellently developed and with very promising results to be able to continue this path towards a treatment that could potentially be applied in the future.
Author Response
Reviewer #2
I have read with great interest the article "Novel Therapeutic Target Critical for SARS-CoV-2 Infectivity and Induction of the Cytokine Release Syndrome". I have found a well thought out idea, excellently developed and with very promising results to be able to continue this path towards a treatment that could potentially be applied in the future. Author response: Thank you for the comment.
Reviewer 3 Report
Line 15: add the word “cells” after “epithelial”
Please write the full time of TLR only one time and then use the abbreviation
Plese write the full name of MMP 9
Please support the methodology part with more references
Line 112: please add a specification to the type of LPS (ultrapure ?)
The concentration of 50 µg/mL LPS seems to be high for a cell stimulation assy. Did you try other concentrations?
Line 159: can you include the name of the alkaline phosphatase substrate in the QUANTI-Blue solution?
Line 181: how did you identify live cells?
Line 198: do you mean with “enzyme-linked antibodies” biotinylated antibodies?
Line 200: I think there must be a washing step between streptavidin-HRP and TMB
Did you use a stop reagent like sulphuric acid?
Do you believe that the Neu-1 could play any role in the MERS-CoV pathogenesis?
As Neu-1 is involved in several cellular activities, could you include possible adverse effects of treatments that target the Neu-1 enzyme?.
The figures are fine in general. The quality of the cartoon figures could be improved.
Author Response
Reviewer #3
1. Line 15: add the word “cells” after “epithelial”: Done
Please write the full time of TLR only one time and then use the abbreviation: Done
Please write the full name of MMP 9: Done
Please support the methodology part with more references. Done
Line 112: please add a specification to the type of LPS (ultrapure ?)The concentration of 50 µg/mL LPS seems to be high for a cell stimulation assay. Did you try other concentrations?: Thank you for this comment and made the correction. "TLR4 ligand, lipopolysaccharide (LPS, from Serratia marcescens, purified by phenol extraction, impurities <3% protein (Lowry) Sigma-Aldrich Canada Co. (Oakville, ON L6H 6J8, Canada) was used at 5 μg/mL as previously reported by us [17]." We have corrected the concentration of LPS throughout the manuscript .
2. Line 159: can you include the name of the alkaline phosphatase substrate in the QUANTI-Blue solution? Done
Line 181: how did you identify live cells? Yes, The kinetic expression of ACE2 (red) and Neu-1 (green) colocalization were taken every second and recorded as a video.
3. Line 198: do you mean with “enzyme-linked antibodies” biotinylated antibodies? Yes, biotinylated antobodies - correction mde.
Line 200: I think there must be a washing step between streptavidin-HRP and TMB Yes, we added this wash step.
Did you use a stop reagent like sulphuric acid? Yes
4. Do you believe that the Neu-1 could play any role in the MERS-CoV pathogenesis? Thank you for the comment: Like SARS-CoV-2, SARS-CoV-1 and MERS-CoV infects human cells through the interaction of the S protein with the ACE2 receptor [46].
5. As Neu-1 is involved in several cellular activities, could you include possible adverse effects of treatments that target the Neu-1 enzyme?. We have included the following in the discussion at the end.
Side Effects:Abdominal or stomach cramps or tenderness.arm, back, or jaw pain.bloating.chest pain or discomfort.diarrhea, watery and severe, which may also be bloody.drooling.facial swelling.fast or irregular heartbeat. What is one serious side effect of Tamiflu? Some people taking Tamiflu have had a severe, life threatening reaction called Stevens-Johnson syndrome. With this condition, a serious rash develops that involves large areas of blisters on your skin. If you have a skin rash while taking Tamiflu, stop taking the drug and call your doctor right away.